# SignKD: Multi-modal Hierarchical Knowledge Distillation for Continuous Sign Language Recognition

## Abstract

Continuous sign language recognition (CSLR) plays a crucial role in promoting inclusivity and facilitating communication within the hearing-impaired community. One of the key challenges in CSLR is accurately capturing the intricate hand movements involved. To address this challenge, we propose a multi-modal framework that first combines video, keypoints, and optical flow modalities to extract more representative features. We investigate various fusion techniques to effectively integrate the information from these modalities. Furthermore, we introduce a hierarchical knowledge distillation (HKD) framework to alleviate the computational burden associated with extracting keypoints and optical flow information. This framework enables the hierarchical transfer of knowledge from multiple modalities to a single-modal CSLR model, ensuring high performance while reducing computational costs. To evaluate the effectiveness of our approach, we conduct extensive experiments on three benchmark datasets: Phoenix-2014, Phoenix-2014T, and CSL-Daily. The results demonstrate that our approach achieves state-of-the-art performance in CSLR, both in the single-stream and multi-stream settings.

## 1 Introduction

Sign language (Tamura & Kawasaki, 1988; Starner et al., 1998) serves as a highly inclusive means of communication for the hearing-impaired community, effectively bridging the gap between deaf and hearing individuals (Zhang et al., 2023). Consequently, the automatic recognition of sign language has emerged as a crucial area of research within the fields of computer vision and artificial intelligence (Cui et al., 2019; Zhou et al., 2021b; Yao et al., 2023; Wei & Chen, 2023). This research direction aims to develop technologies and systems that can accurately interpret and understand sign language, thereby facilitating effective communication and enhancing accessibility for individuals with hearing impairments. Sign Language Recognition can be divided into two primary categories: isolated sign language recognition (ISLR) (Sincan et al., 2021; Vázquez-Enríquez et al., 2021; Lee et al., 2023; Laines et al., 2023) and continuous sign language recognition (CSLR) (Chen et al., 2022b; Joze & Koller, 2018; Tang et al., 2021). CSLR poses a greater challenge than ISLR, as the latter focuses on the classification of isolated sign videos into single gloss, while CSLR involves the intricate task of transcribing co-articulated sign videos into sign sequences on a gloss-by-gloss basis. Since ISLR fails to capture the continuous and context-dependent nature of sign language, a more natural and accurate approach is required to recognize sign language in a continuous fashion and understand the complex temporal dependencies between signs.

Recently, deep learning models have shown remarkable success in various computer vision tasks, including SLR (Hu et al., 2023a; Zhou et al., 2023; Zuo et al., 2023). Many approaches (Cihan Camgoz et al., 2017; Cheng et al., 2020; Chen et al., 2022a) for SLR rely on computer vision techniques that extract features solely from video sequences. However, these methods often encounter challenges in effectively addressing the inherent complexities of sign language, such as the high dimensionality of the data, dynamic hand movements, and variations in signing styles among individuals. Additionally, the lack of large-scale, finely-annotated sign datasets in the field of SLR often leads to insufficient training in these models.

Therefore, numerous studies (Koller et al., 2019; Zuo & Mak, 2022; Chen et al., 2022b) focused on developing multi-stream solutions, i.e., video-flow or video-keypoints dual-stream networks. The multi-stream architectures extend multi-cue visual features, yielding the current state-of-the-art performances. However, previous methods fail to aggregate the three modalities, i.e., video, keypoints, and optical flow, to utilize the potential connection among them as an enhancement for CSLR. Furthermore, they face the problem of large computation costs and low efficiency. As a result, achieving high sign recognition accuracy with efficient computational performance in real-world scenarios remains a formidable task.

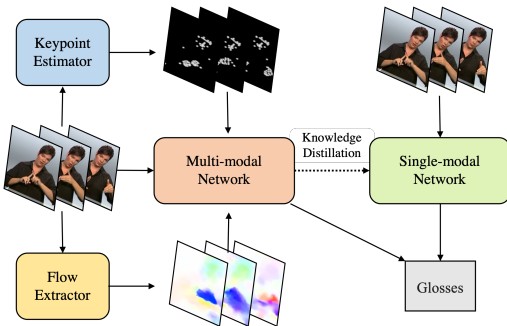

Figure 1: **Our SignKD architecture.** We train a multi-modal network using video, keypoints and optical flow as the teacher model. And then use our proposed hierarchical knowledge distillation framework to distill the multi-modal information to the single-modal network which only use video as input.

To address the challenge of mentioned above, we propose a novel methodology that combines video, keypoints, and optical flow modalities. Additionally, we investigate various fusion methods to effectively integrate the information from these modalities, including MLP-based fusion, attention-based fusion and convolution-based fusion. To mitigate the computational overhead associated with extracting keypoints and optical flow information, we introduce a Hierarchical Knowledge Distillation framework. This framework enables the transfer of knowledge from multiple modalities to a single-modal SLR model in a hierarchical manner. By leveraging this approach, we can maintain the high performance as the larger model while reducing the computational cost and resource requirements. Figure 1 describes the central concept of this work.

To validate the effectiveness of our proposed method, we conduct extensive experiments on Phoenix-2014 (Koller et al., 2015), Phoenix-2014T (Camgoz et al., 2018) and CSL-Daily (Zhou et al., 2021a). The results demonstrate that our approach achieves state-of-the-art performance in continuous sign language recognition in both single-modal settings and multi-modal settings.

Our main contributions can be summarized as follows:

- We are the **first** to effectively utilize video, keypoints, and optical flow modalities together to capture dynamic hand movement information in sign language recognition.
- We present a hierarchical knowledge distillation framework that enhances the accuracy of single-modal model while taking lower computational costs than multi-modal model.
- Our results demonstrate significant improvements achieved by SignKD across multiple benchmarks in both multi-modal and single-modal settings.

## 2 RELATED WORKS

**Sign language recognition.** Early efforts in CSLR primarily relied on hand-crafted features (Han et al., 2009; Koller et al., 2015) or Hidden Markov Model-based systems (Koller et al., 2016; 2017). These approaches, while foundational, struggled to capture the inherent complexity and variability of sign language. In recent years, deep learning-based methodologies have ushered in a paradigm shift in the domain of CSLR, which can be succinctly encapsulated within three pivotal phases: feature extraction, recognition, and alignment. Predominantly, 3D CNNs (Pu et al., 2019; Li et al., 2020; Chen et al., 2022a;b; Joze & Koller, 2018) have gained widespread adoption for feature extraction. Additionally, certain approaches (Hu et al., 2023b; Min et al., 2021; Zhou et al., 2021b; Cui et al., 2019) opt to commence with a 2D CNN to extract frame-wise features before subsequently incorporating hybrid architectures composed of 1D CNNs and Long Short-Term Memory (LSTM) networks to capture temporal dependencies. Upon deriving features, classifiers can compute posterior probabilities to facilitate the recognition process. Since CSLR is a weakly supervised sequence-to-sequence task without temporal boundary annotation available, CTC loss (Graves et al., 2006) is widely used to find the proper alignment between clips and glosses to ensure an accurate training procedure.

**Kepoints and optical flow in action recognition.** Keypoints play a crucial role in action recognition, as they provide valuable information about the spatial and temporal characteristics of actions in videos or image sequences. Various CNN and RNN based methods (Zhang et al., 2019b; Zhao et al., 2019; Li et al., 2018; Hernandez Ruiz et al., 2017) have been proposed for kepoints action recognition. Commencing with ST-GCN (Yan et al., 2018), a succession of keypoint-based approaches (Chen et al., 2021b;c; Chi et al., 2022) has leveraged Graph Convolutional Networks (GCNs) to effectively model spatio-temporal relationships, consistently delivering superior performance results. Leveraging keypoints to enhance the performance of SLR remains an ongoing challenge. Recent research endeavors (Zhou et al., 2021b; Tang et al., 2021; Chen et al., 2022b) have proposed innovative methodologies and made progress in this regard. Optical flow provides information about how pixels in consecutive frames of a video are moving. Consequently, numerous studies (Feichtenhofer et al., 2016; Simonyan & Zisserman, 2014; Wang et al., 2019; Sun et al., 2018) have adopted the strategy of decomposing videos into spatial and temporal components through the utilization of RGB and optical flow frames, ultimately yielding state-of-the-art results in the realm of action recognition. In a similar vein, within the domain of SLR, Cui et al. (2019) have delved into the multi-modal fusion of RGB images and optical flow data as an approach to enhance the recognition accuracy.

**Knowledge distillation.** Knowledge distillation technology enables the extraction of knowledge from a teacher model to guide the training of student models. Leveraging this technology, we adopt a similar approach by utilizing the multi-modal features learned by a multi-modal model to guide the training of a single-modal model. This allows us to obtain an efficient and high-performance student model specifically designed for SLR. Initially, Breiman & Shang (1996) pioneered the concept of learning singletree models, which approximate the performance of multipletree models and offer enhanced interpretability. In the context of neural networks, similar approaches emerged through the work of Buciluǎ et al. (2006), Ba & Caruana (2014), and Hinton et al. (2015), primarily aimed at model compression. More recently, Zhang et al. (2019a), on the other hand, directly employed the final pose maps of teacher networks to supervise both the interim and final results of student networks. Additionally, He et al. (2019) and Liu et al. (2019) have focused on distilling knowledge embedded within layer-wise features for tasks such as semantic segmentation. Liu et al. (2020) introduced a structured knowledge transfer framework that facilitates the transfer of knowledge from teacher networks to student networks, encompassing both inter-part and intra-part aspects. Zhao et al. (2022) presented an effective logit distillation method named decoupled knowledge distillation. Knowledge distillation has seen broad utilization in numerous fields. In this work, we propose a hierarchical knowledge distillation method to distill knowledge from multiple modalities into a single-modal SLR.

## 3 METHODS

In this section, we present our SignKD approach for continuous sign language recognition. Firstly, we introduce our multi-modal feature extractor, which is the **first** to combine video, keypoints, and optical flow to effectively capture dynamic hand movement information. Next, we discuss our proposed multi-modal feature fusion module and various fusion strategies for integrating the features from multiple modalities. Lastly, we propose a hierarchical knowledge distillation framework that transfers knowledge from multiple modalities to a single-modal SLR model in a hierarchical manner. This ensures the effective distillation of knowledge across different levels. Figure 2 illustrates the design of our framework.

### 3.1 MULTI-MODAL FEATURE EXTRACTOR

In this subsection, we provide individual introductions for each of the three streams that compose the feature encoders of our multi-modal model. By incorporating video, keypoints and optical flow, our model can effectively capture movement patterns and hand gestures, which are crucial in sign language understanding.

**Video encoder.** We adopt S3D (Xie et al., 2018) with a lightweight head network as our video encoder. However, to achieve dense temporal representations, we only utilize the first four blocks of S3D. Each input video of dimensions $T \times H \times W \times 3$ is fed into the encoder to extract its features, where T denotes the frame number and H and W represent the height and width of the video by default. We pool the output feature of the last S3D block into $T/4 \times 832$ before feeding it into the

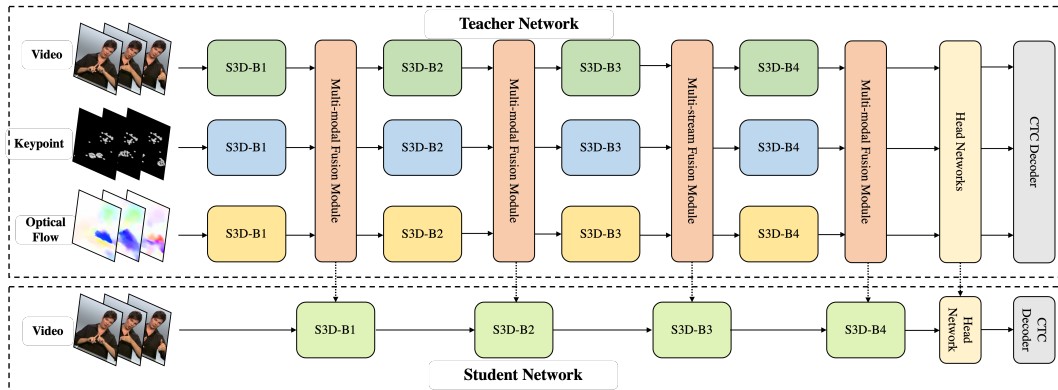

Figure 2: **Illustration of our framework.** Our multi-modal teacher network use video, keypoints and optical flow as inputs and single-modal student network only utilizes video as input. For the multi-modal network, we use S3D for feature representation learning, a feature fusion module to fuse the feature of different modalities, and head networks and CTC decoder to decoder the features to glosses. The components are the same for the student network except for the fusion module.

head network, which aims to capture further temporal context. Then the feature is feed to the video head network, which consists of a temporal linear layer, a batch normalization layer, a ReLU layer, and a temporal convolutional block with two temporal convolutional layers with a kernel size of 3 and stride 1, a linear translation layer, and another ReLU layer. The output feature is named as gloss representation with a size of $T/4 \times 512$. Frame-level gloss probabilities are extracted using a linear classifier and Softmax function. Finally, we optimize the video encoder using connectionist temporal classification (CTC) loss (Graves et al., 2006).

**Keypoint encoder.** We adopt the keypoints generated by HRNet (Wang et al., 2020) trained on COCO-WholeBody (Jin et al., 2020), which contains 42 hand keypoints, 26 face keypoints covering the mouth, eyes, and face contour, and 11 upper body keypoints covering shoulders, elbows, and wrists per frame, to model the keypoint sequences. In total, 79 keypoints are employed, and we represent the keypoints using heatmaps. Concretely, denoting the keypoint heatmap sequence with a size of $T \times H' \times W' \times K$ as $G$, where $H'$ and $W'$ represent the spatial resolution of each heatmap and $K$ is the total number of keypoints, the elements of $G$ are computed using a Gaussian function represents the coordinates of the $k - th$ keypoint in the $t - th$ frame and $\sigma$ controls the scale of keypoints. We set $\sigma = 4$ by default and $H' = W' = 112$ to reduce computational cost. The network architecture of the keypoint encoder is the same as the video encoder except for the first convolutional layer, which is modified to accommodate keypoints input. The weights of the video encoder and keypoint encoder are not shared to extract the representative features of each modality. Similarly, a CTC loss is used to train the keypoint encoder.

**Optical flow encoder.** The optical flow encoder plays a crucial role in our multi-modal framework. We adopt dense optical flow extracted by TV-L1 algorithm (Sánchez Pérez et al., 2013) as the source of motion representation. The network architecture of the optical flow encoder is similar to that of the video encoder and keypoint encoder, except for the first layer. The utilization of optical flow as a representation for sign language recognition serves several important purposes. Firstly, optical flow provides a dense motion representation, capturing the spatial-temporal dynamics inherent in sign language. By calculating the displacement of pixels between consecutive frames, optical flow encodes the motion information that is essential for understanding sign language gestures. This enables the model to capture finer details of the signing motion and facilitates accurate recognition. Secondly, employing optical flow as an intermediate encoding enables the network to focus on capturing the movement patterns and hand gestures, effectively bypassing irrelevant visual information such as static backgrounds or lighting variations. This enhances the discriminative power of the network by emphasizing the most relevant features for SLR. Furthermore, by incorporating optical flow in our framework, we exploit the complementary nature of appearance-based features from the video encoder and motion-based features from the optical flow encoder and keypoint encoder. The fusion of these modalities allows our model to capture both static and dynamic aspects of sign language, enabling a more comprehensive understanding of the signing gestures.

## 3.2 Multi-modal network

**Multi-modal fusion.** The multi-modal feature fusion module is a critical component in our framework for continuous sign language recognition. It aims to integrate the information from the video, keypoints, and optical flow modality, leveraging their complementary characteristics to enhance the model's discriminative power.

By combining the feature representations from the video encoder, keypoint encoder, and optical flow encoder, the feature fusion module enables a more comprehensive representation of sign language. Through advanced fusion techniques like MLP-based fusion, attention-based fusion, and convolution-based fusion, the module effectively merges features at different abstraction levels. This fusion module plays a vital role in capturing a broader range of cues by leveraging the complementary nature of the different modalities. The integration of multiple modalities allows the model to incorporate appearance, motion, and spatial-temporal information, resulting in more informative and discriminative features. Ultimately, this leads to improved accuracy in recognizing sign language gestures. In our multi-modal model, the multi-modal feature fusion module acts as a bridge, consolidating the outputs from the video, keypoints, and optical flow encoders. This integration enables a more comprehensive understanding of sign language by considering various modalities simultaneously. By leveraging the strengths of each modality, our multi-modal model achieves enhanced performance in continuous sign language recognition.

**Different fusion strategy.** In our framework, we explore different fusion methods within the multi-modal feature fusion module, including MLP-based, attention-based, and convolution-based methods. These methods effectively combine the feature representations from the video, keypoints, and optical flow modalities. We conducted ablation studies to analyze the performance of each fusion method and found that MLP-based method yielded superior results. Please see Section B for the details.

MLP-based method allow for flexible modeling of the fusion process, capturing complex relationships among the input features. While attention-based method (Huang et al., 2019; Chen et al., 2021a) and convolution-based method (Liu et al., 2021) have shown success in various tasks, our experiments revealed that they were not as effective as MLP-based method for sign language recognition. The attention mechanism aims to emphasize relevant features by assigning weights to different parts of the input, while convolution-based method focuses more on learning between local information of different modalities. However, in our specific context of continuous sign language recognition, MLP-based fusion demonstrated superior performance. The superior performance of MLP-based fusion can be attributed to its ability to capture intricate relationships between features from different modalities, enabling effective integration of complementary information. This finding highlights the importance of exploring and evaluating different fusion techniques tailored to the specific requirements of sign language recognition tasks.

**Multi-modal SLR loss.** The total loss of our multi-modal SLR consists of three parts. First, the CTC losses applied on the outputs of the video encoder $L_{CTC}^V$, keypoint encoder $L_{CTC}^K$, optical flow encoder $L_{CTC}^{OF}$ and joint head encoder $L_{CTC}^J$. Second, the auxiliary CTC losses $L_{SPN}$ applied on the outputs of our sign pyramid networks. The third is the self-distillation loss $L_{SD}$ which is implemented by the Kullback-Leibler divergence (Kullback & Leibler, 1951) loss. Please refer to the section C for the details of our joint head, sign pyramid networks, and self-distillation design. The total loss for training the multi-modal SLR $L_{SLR}^T$ can be calculated as follows:

$$L_{SLR}^T = \lambda_{CTC}(L_{CTC}^V + L_{CTC}^K + L_{CTC}^{OF} + L_{CTC}^J) + \lambda_{SPN}L_{SPN} + \lambda_{SD}L_{SD}, \tag{1}$$

where $\lambda_{CTC}$, $\lambda_{SPN}$, and $\lambda_{SD}$ are loss weights. Once the training process is completed, the multi-modal SLR model becomes capable of predicting a gloss sequence. This is achieved by averaging the predictions obtained from the four head networks.

## 3.3 Multi-modal Hierarchical Knowledge Distillation

To address the challenge of cross-modal knowledge transfer in CSLR, we propose a novel method called Hierarchical Knowledge Distillation (HKD). Our method focuses on distilling knowledge from multiple modalities into a single-modal SLR model using knowledge distillation (KD) technique. Specifically, the teacher network in our approach is a multi-modal SLR model, which incor-

porates video, keypoints, and optical flow modalities. The student network, on the other hand, is a single-modal SLR model that utilizes only the video modality as input.

To perform knowledge distillation, we employ a hierarchical approach in HKD. This involves transferring knowledge from the shallow layers to the deep layers of the network. The hierarchical distillation process ensures that the student network learns from the fused features at various levels of abstraction, capturing both low-level details and high-level semantic information.

The adoption of HKD offers several advantages in CSLR. Firstly, it improves the inference speed of the multi-modal SLR model. By distilling knowledge from multiple modalities into a unified representation, HKD eliminates the need for extracting optical flow and keypoints information during testing, reducing the computational complexity and accelerating the inference process. Secondly, HKD enhances the accuracy of the single-modal SLR model. By distilling knowledge hierarchically from the MLP-fused features, the student network benefits from the discriminative power and contextual information captured by the teacher network. This integration of diverse cues, such as appearance, motion, and spatial-temporal information, leads to improved recognition accuracy and better discrimination of sign language gestures, which makes our method more efficient and effective in real-world applications.

**Hierarchical knowledge distillation loss.** The hierarchical knowledge distillation loss in our approach comprises three components. The first component is the hierarchical feature loss, denoted as $L_F$. In this step, the fused multiple-modal features are passed through a convolutional layer. The cosine loss is then calculated between these fused features and the single-modal features. This loss encourages consistency and alignment between the features extracted from multiple modalities and single modality. For details, the feature groups $T$ and $S$ selected from teacher model and student model can be represented as:

$$T = \{f_t^1, f_t^2, f_t^3, f_t^4\}, S = \{f_s^1, f_s^2, f_s^3, f_s^4\}, \tag{2}$$

where $f_t^i$ represent the features selected from each stage of our multi-modal fusion module and $f_s^i$ are features output from each block of our video modality modal. We generate a group of interim features $H = \{f_h^1, f_h^2, f_h^3, f_h^4\}$ by feeding each feature $f_s^i$ in $S$ into a $3 \times 3 \times 3$ convolutional layer. Then the similarity of $f_t^i$ and $f_h^i$ at location $(x, y, t)$ can be calculated by:

$$Cos\{f_h^i(x,y,t), f_t^i(x,y,t)\} = \sum_{c=1}^{C^i} \frac{f_h^i(x,y,t,c) \cdot f_t^i(x,y,t,c)}{|f_h^i(x,y,t)| \cdot |f_t^i(x,y,t)|}, \tag{3}$$

where $C^i$ denotes the channel number of feature $f_t^i$, $f_t^i(x,y,t,c)$ is the response value of $f_t^i$ at location $(x, y, t)$ of the $c$-th channel. Finally, $L_F$ is defined as follows

$$L_F = \sum_{f_t^i, f_h^i} \sum_{x=1}^{H^i} \sum_{y=1}^{W^i} \sum_{t=1}^{T^i} 1 - Cos\{f_h^i(x,y,t), f_t^i(x,y,t)\}, \tag{4}$$

where $H^i$, $W^i$ and $T^i$ are the height, width and frames of feature $f_t^i$.

The second component is the Kullback-Leibler divergence loss, denoted as $L_O$, which measures the divergence between the output probabilities of the teacher network and the student network. This loss guides the student network to mimic the output distribution of the teacher network, promoting knowledge transfer. The third component is the CTC loss applied to the outputs of the video decoder of the student network, denoted as $L_{CTC}^{VS}$. This loss function helps align the predicted sequence from the student network with the ground truth labels, aiding accurate recognition. The hierarchical knowledge distillation loss, denoted as $L_{HKD}$, is formulated by combining these three components as follows:

$$L_{HKD} = \lambda_F L_F + \lambda_O L_O + \lambda_{CTC\_VS} L_{CTC}^{VS}, \tag{5}$$

where $\lambda_F$, $\lambda_O$, and $\lambda_{CTC\_VS}$ are loss weights and set to 0.5, 1.0 and 1.0, respectively.

## 4 EXPERIMENTS

### 4.1 EXPERIMENTAL SETUP

**Datasets.** *Phoenix-2014* (Koller et al., 2015) and *Phoenix-2014T* (Camgoz et al., 2018) are two German sign language (DGS) datasets widely used in the field of continous sign language translation

Table 1: **Comparison with previous works on the Phoenix-2014 and Phoenix-2014T datasets.** The best results and previous best results are marked as **bold** and underlined.

| Method | Modality | Phoenix-2014 | | Phoenix-2014T | |
|---|---|---|---|---|---|
| | | Dev | Test | Dev | Test |
| DNF (Cui et al., 2019) | video+flow | 23.1 | 22.9 | - | - |
| STMC-R (Zhou et al., 2021b) | video+keypoints | 21.1 | 20.7 | 19.6 | 21.0 |
| C$^2$SLR (Zuo & Mak, 2022) | video+keypoints | 20.5 | 20.4 | 20.2 | 20.4 |
| TwoStream-SLR (Chen et al., 2022b) | video+keypoints | 18.4 | 18.8 | 17.7 | 19.3 |
| Ours | video+keypoints+flow | **17.1** | **17.3** | **16.5** | **18.1** |
| SubUNets (Cihan Camgoz et al., 2017) | video | 40.8 | 40.7 | - | - |
| SFL (Niu & Mak, 2020) | video | 23.8 | 24.4 | - | - |
| FCN (Cheng et al., 2020) | video | 23.7 | 23.9 | 23.3 | 25.1 |
| Joint-SLRT (Camgoz et al., 2020) | video | - | - | 24.6 | 24.5 |
| VAC (Min et al., 2021) | video | 21.2 | 22.3 | - | - |
| SignBT (Zhou et al., 2021a) | video | - | - | 22.7 | 23.9 |
| SMKD (Hao et al., 2021) | video | 20.8 | 21.0 | 20.8 | 22.4 |
| MMTLB (Chen et al., 2022a) | video | - | - | 21.9 | 22.5 |
| TLP (Hu et al., 2022) | video | 19.7 | 20.8 | 19.4 | 21.2 |
| CorrNet (Hu et al., 2023b) | video | 18.8 | 19.4 | 18.9 | 20.5 |
| CVT-SLR (Zheng et al., 2023) | video | 19.8 | 20.1 | 19.4 | 20.3 |
| SEN (Hu et al., 2023c) | video | 19.5 | 21.0 | 19.3 | 20.7 |
| Ours | video | **18.5** | **18.9** | **18.3** | **19.9** |

(CSLR). The Phoenix-2014 dataset consists of 5672 training, 540 development, and 629 testing samples, with a vocabulary size of 1081 for glosses (sign language labels). It covers a total of 1231 unique signs. On the other hand, Phoenix-2014T is an extension of Phoenix-2014, containing 7096 training, 519 development, and 642 testing samples. It has a larger vocabulary size of 1066 for glosses and 2887 for German text, which includes translations transcribed from the news speaker. The two datasets share 958 common signs, providing a valuable resource for evaluating SLR models.

*CSL-Daily* (Zhou et al., 2021a) is a recently released large-scale Chinese sign language (CSL) dataset specifically designed for sign language translation tasks. It comprises 18401 training, 1077 development, and 1176 testing video segments. The dataset captures performances of ten different signers and covers various topics such as family life, medical care, and school life. CSL-Daily features a vocabulary size of 2000 for glosses and 2343 for Chinese text. It serves as a valuable benchmark for evaluating the performance of SLR models in the context of Chinese sign language.

**Evaluation metrics.** The Word Error Rate (WER) stands as the predominant metric for assessing the performance of Sign Language Recognition (SLR). It quantifies the essential insertions (#ins), substitutions (#sub), and deletions (#del) required to align recognized sentences with their corresponding reference sentences (#reference). The lower WER, the better accuracy.

$$\text{WER} = \frac{\#ins + \#sub + \#del}{\#reference}. \tag{6}$$

**Implementation details.** In both the Phoenix-2014 and Phoenix-2014T datasets, data from three modalities are cropped to dimensions of $224 \times 224$, while in the CSL-Daily dataset, a crop size of $320 \times 320$ is applied. During the training phase, data augmentations are applied, consisting of spatial cropping within the range of [0.7-1.0] and frame-rate augmentation spanning [×0.5-×1.5]. For our network training strategy, we implement a cosine annealing schedule spanning 60 epochs. We utilize the Adam optimizer with a weight decay of $1e^{-3}$ and set the initial learning rate to $1e^{-3}$. We train our models on 4 Nvidia A100 GPUs. In the inference stage, we employ a CTC decoder to derive the final gloss predictions. Specifically, a beam width of 5 is configured for the decoding.

Table 2: **Comparison with previous works on the CSL-Daily dataset.** The best results and previous best results are marked as **bold** and underlined.

| Method | Modality | CSL-Daily | |
|---|---|---|---|
| | | Dev | Test |
| TwoStream-SLR (Chen et al., 2022b) | video+keypoints | 25.4 | 25.3 |
| Ours | video+keypoints+flow | **24.3** | **24.0** |
| SubUNets (Cihan Camgoz et al., 2017) | video | 41.4 | 41.0 |
| LS-HAN (Huang et al., 2018) | video | 39.0 | 39.4 |
| FCN (Cheng et al., 2020) | video | 33.2 | 33.5 |
| Joint-SLRT (Camgoz et al., 2020) | video | 33.1 | 32.0 |
| SignBT (Zhou et al., 2021a) | video | 33.2 | 33.2 |
| CorrNet (Hu et al., 2023b) | video | 30.6 | 30.1 |
| SEN (Hu et al., 2023c) | video | 31.1 | 30.7 |
| Ours | video | **26.8** | **26.6** |

## 4.2 COMPARISON WITH STATE-OF-THE-ART METHODS

As illustrated in Table 1, we conducted a comparative analysis between our proposed multi-modal model, which integrates video, keypoints, and optical flow modalities, and the existing state-of-the-art methods. Our approach outperformed all other methods, establishing new state-of-the-art results on both the Phoenix-2014 and Phoenix-2014T datasets. The optimal WER scores achieved by our multi-modal model on the testing set outperforms the previous best method by 7.9% on Phoenix-2014 and 6.2% on Phoenix-2014T, respectively. Our single-modal model also achieves the lowest WER, which are 18.5% and 18.9% on the Phoenix-2014 dataset, and 18.3% and 19.9% on the Phoenix-2014T dataset, respectively. In Table 2, we present a comparative analysis between our multi-modal model and the previous state-of-the-art method TwoStream-SLR on the CSL-Daily dataset. Our model exhibits a notable reduction in WER by 1.1% on the development set and 1.3% on the testing set when compared to TwoStream-SLR, showcasing its superior performance. The performance of our single-modal model significantly surpasses that of other models relying solely on video data. Our model achieves a substantial reduction in word error rates, outperforming the previous best models by 12.4% on the development set and 11.6% on the testing set.

## 4.3 ABLATION STUDIES

**Effect of each encoder in our multi-modal model.** In table 3, we assessed the performance of sign language recognition by individually employing the video encoder, keypoint encoder, and flow encoder. Notably, the video encoder yielded the most favorable outcomes, while the flow encoder demonstrated comparatively less effectiveness. Specifically, on the development and testing sets, WER of 21.2% and 21.8%, respectively, were achieved using the video encoder, while the flow encoder resulted in WER of 32.1% and 30.8%, respectively. Upon combining these three encoders, a substantial enhancement in sign language recognition performance was observed, showcasing a significant reduction in WER compared to using any single encoder in isolation.

**Study on fusion methods.** We also investigate the efficacy of our four-stage fusion module in Table 3. It became evident that each fusion stage contributed to a reduction in sign language recognition error rates. When all four stages of the fusion module were employed in tandem, a notable improvement was observed, resulting in WER reductions of 3.0% on the development set and 2.2% on the testing set, as compared to scenarios where the fusion module was not utilized. Finally, we conducted a comprehensive study of various fusion methods. Our findings revealed that the MLP-based fusion method outperformed both the attention-based and convolution-based methods, underscoring its superior effectiveness in enhancing sign language recognition.

**Study on training losses.** In order to encompass glosses spanning various temporal extents and encourage intermediate layers to acquire more semantically meaningful features, we leverage a Sign Pyramid Network (SPN) to craft auxiliary CTC losses. Additionally, we employ averaged gloss probabilities from four heads as pseudo-targets for fine-grained supervision, creating a self-

Table 3: **Ablation study of each component of our multi-modal model on the Phoenix-2014T.** 'VE', 'KE', 'FE' are video encoder, keypoint encoder and flow encoder, respectively. 'M1' $\sim$ 'M4' represent the four fusion stages within our multi-modal model. 'MLP', 'Attention' and 'Convolution' are different fusion methods.

| VE | KE | FE | M1 | M2 | M3 | M4 | MLP | Attention | Convolution | Dev | Test |
|----|----|----|----|----|----|----|-----|-----------|-------------|-----|------|
| ✓ |   |   |   |   |   |   |   |   |   | 21.2 | 21.8 |
|   | ✓ |   |   |   |   |   |   |   |   | 26.0 | 25.2 |
|   |   | ✓ |   |   |   |   |   |   |   | 32.1 | 30.8 |
| ✓ | ✓ |   |   |   |   |   |   |   |   | 20.4 | 21.0 |
| ✓ | ✓ | ✓ |   |   |   |   |   |   |   | 19.5 | 20.3 |
| ✓ | ✓ | ✓ | ✓ |   |   |   | ✓ |   |   | 18.3 | 19.5 |
| ✓ | ✓ | ✓ | ✓ | ✓ |   |   | ✓ |   |   | 17.6 | 18.9 |
| ✓ | ✓ | ✓ | ✓ | ✓ | ✓ |   | ✓ |   |   | 17.0 | 18.5 |
| ✓ | ✓ | ✓ | ✓ | ✓ | ✓ | ✓ | ✓ |   |   | **16.5** | **18.1** |
| ✓ | ✓ | ✓ | ✓ | ✓ | ✓ | ✓ |   | ✓ |   | 17.0 | 18.8 |
| ✓ | ✓ | ✓ | ✓ | ✓ | ✓ | ✓ |   |   | ✓ | 17.4 | 19.0 |

Table 4: **Effect of each loss used in training the multi-modal model and single-modal model on the Phoenix-2014T.** In subtable (b), 'CTC' means we use CTC loss to train single-modal model, 'S1' $\sim$ 'S4' means we use feature distillation loss in each stage, 'O' means the output distillation loss.

(a) Weights of loss in multi-modal model.

| $\lambda_{CTC}$ | $\lambda_{SPN}$ | $\lambda_{SD}$ | Dev | Test |
|-----------------|-----------------|----------------|-----|------|
| 1.0 |   |   | 17.4 | 19.0 |
| 1.0 | 0.2 |   | 16.9 | 18.6 |
| 1.0 | 0.5 |   | 17.1 | 18.9 |
| 1.0 | 0.2 | 0.5 | 16.6 | 18.3 |
| 1.0 | 0.2 | 1.0 | **16.5** | **18.1** |

(b) Effect of losses in single-modal model.

| $CTC$ | $S1$ | $S2$ | $S3$ | $S4$ | $O$ | Dev | Test |
|-------|------|------|------|------|-----|-----|------|
| ✓ |   |   |   |   |   | 21.2 | 21.8 |
| ✓ | ✓ |   |   |   |   | 20.5 | 21.3 |
| ✓ | ✓ | ✓ |   |   |   | 20.0 | 21.0 |
| ✓ | ✓ | ✓ | ✓ |   |   | 19.4 | 20.7 |
| ✓ | ✓ | ✓ | ✓ | ✓ |   | 19.0 | 20.3 |
| ✓ | ✓ | ✓ | ✓ | ✓ | ✓ | **18.3** | **19.9** |

distillation loss. In Table 4(a), while maintaining a fixed CTC loss weight of 1.0, we observe that setting $\lambda_{SPN}$ to 0.2 and $\lambda_{SD}$ to 1.0 yields the optimal performance.

In Table 4(b), our investigation centers on the impact of distillation loss during the training of the single-modal model. While maintaining a constant CTC loss weight of 1.0, we set the losses for distillation of latent features to a uniform value of 0.1 and assign a weight of 1.0 to the loss for distillation of output gloss probabilities. Notably, when these loss components are combined and used to train the model simultaneously, we observe a significant reduction in WER by 13.7% on the development set and 8.7% on the test set.

## 5 CONCLUSION

In this paper, we addressed the challenge of continuous sign language recognition (CSLR) by proposing SignKD, a novel framework that combines video, keypoints, and optical flow modalities. We introduced a hierarchical knowledge distillation approach to transfer knowledge from multiple modalities to a single-modal CSLR model. Our methodology effectively captured the intricate hand movements inherent in sign language and reduced the computational overhead associated with extracting keypoints and optical flow information. Through extensive experiments on Phoenix-2014, Phoenix-2014T, and CSL-Daily datasets, we demonstrated the effectiveness of SignKD in achieving state-of-the-art performance in CSLR. Our results showed significant improvements in both single-stream and multi-stream settings, highlighting the advantages of our approach.

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

## A  OVERVIEW

- Section B: Details of fusion methods.
- Section C: More designs in multi-modal model.

## B  DETAILS OF FUSION METHODS.

For the MLP-based fusion method, we use a simple but effective method, which is a MLP containing three linear layers, each layer followed by a GELU activation function except for the last one. The operation of fusion module can be represented as follow:

$$F_{fuse} = \mathfrak{R}_{\mathfrak{m}}(F_{video} \copyright F_{keypoint} \copyright F_{flow}), \tag{7}$$

$$F'_{video} = F_{fuse} + F_{video}, \tag{8}$$

$$F'_{keypoint} = F_{fuse} + F_{keypoint}, \tag{9}$$

$$F'_{flow} = F_{fuse} + F_{flow}, \tag{10}$$

where $\mathfrak{R}_{\mathfrak{m}}$ can represent the MLP function, $F_{fuse}$ is the feature output from our fusion module, $F_{video}$, $F_{keypoint}$, $F_{flow}$ are features input to fusion module, and $F'_{video}$, $F'_{keypoint}$, $F'_{flow}$ are enhanced features input to the next stage. The same meanings of the symbols below.

For the attention-based fusion method, the operation of fusion module can be represented as:

$$F'_{video} = F_{video} + \mathfrak{R}_{\mathfrak{c}}(F_{video}, F_{keypoint}) + \mathfrak{R}_{\mathfrak{c}}(F_{keypoint}, F_{flow}), \tag{11}$$

$$F'_{keypoint} = F_{keypoint} + \mathfrak{R}_{\mathfrak{c}}(F_{keypoint}, F_{video}) + \mathfrak{R}_{\mathfrak{c}}(F_{keypoint}, F_{flow}), \tag{12}$$

$$F'_{flow} = F_{flow} + \mathfrak{R}_{\mathfrak{c}}(F_{flow}, F_{video}) + \mathfrak{R}_{\mathfrak{c}}(F_{flow}, F_{keypoint}), \tag{13}$$

where $\mathfrak{R}_{\mathfrak{c}}$ can represent the cross attention function.

In terms of convolution-based fusion method, we describe the process as:

$$F_{fuse} = \mathfrak{R}_{\mathfrak{1}}(\mathfrak{R}_{\mathfrak{2}}(F_{video} \copyright F_{keypoint} \copyright F_{flow})) \tag{14}$$

$$F'_{video} = F_{fuse} + F_{video}, \tag{15}$$

$$F'_{keypoint} = F_{fuse} + F_{keypoint}, \tag{16}$$

$$F'_{flow} = F_{fuse} + F_{flow}, \tag{17}$$

where $\mathfrak{R}_{\mathfrak{1}}$ and $\mathfrak{R}_{\mathfrak{2}}$ can be $1 \times 1 \times 1$ convolutional layer and $3 \times 3 \times 3$ convolutional layer.

## C  MORE DESIGNS IN MULTI-MODAL MODEL.

We utilize a combined head and late ensemble method to merge the multi-modal features, as shown in Figure 3 (a). This is necessary because each encoder has its own head network. Additionally, we employ signal pyramid networks that effectively capture glosses across various temporal spans, as shown in Figure 3 (b). It also facilitates the supervision of shallow layers, enabling them to learn significant representations. Furthermore, we use a frame-level self-distillation method, which not only offers frame-level supervision but also transfers knowledge from the late ensemble back to each individual modality.

**Joint Head and Late Ensemble.** In our model, we have separate head networks for the video encoder, keypoint encoder, and optical flow encoder. However, to fully exploit the potential of our dual encoder architecture, we introduce an additional head network called the joint head. This joint head takes the concatenation of the outputs from the three S3D networks as inputs. The architecture of the joint head is the same as the individual video head, keypoint head, and flow head. The joint head is also supervised by a CTC loss, ensuring comprehensive training. To generate the final gloss sequence, we employ a late ensemble strategy. We average the frame-wise gloss probabilities predicted by the video head, keypoint head, optical flow head, and joint head. These averaged probabilities are then passed through a CTC decoder, which generates the final gloss sequence. This late ensemble strategy combines the results from multiple streams and improves upon the predictions made by individual streams. By incorporating the joint head and employing the late ensemble

Figure 3: (a) Sign pyramid network use the features of the last three blocks of S3D as the input and feed the features to the head network. (b) The head networks in our mulit-modal model consists of a video head, a keypoint head, a flow head and a joint head to generate gloss probabilities, then we feed the probabilities to a CTC decoder to generate the gloss sequence.

strategy, we maximize the benefits of our multi-stream architecture, leveraging the strengths of each stream and achieving improved performance compared to single-stream predictions.

**Sign Pyramid Network.** To capture glosses of different temporal spans and effectively supervise the shallow layers for meaningful representations, we incorporate a sign pyramid network (SPN) with auxiliary supervision into our model. This approach builds upon previous research and involves a top-down pathway and lateral connections within the SPN. To fuse features extracted by adjacent S3D blocks, we utilize an element-wise addition operation. Additionally, transposed convolutions are employed to match the temporal and spatial dimensions of the two feature maps before performing element-wise addition. Two separate head networks, following the same architecture as the dual encoder, are utilized to extract frame-level gloss probabilities. Auxiliary supervision is provided by employing CTC losses. We employ three independent SPNs for the video, keypoints, and optical flow modalities, ensuring comprehensive coverage across different modalities.

**Self Knowledge distillation.** Due to the lack of labeled temporal boundaries for glosses in existing datasets, CTC loss has been widely used to leverage weak supervision at the sentence level. However, once the visual encoder is well optimized, it becomes capable of generating frame-wise gloss probabilities, which can be used to estimate approximate temporal boundaries for glosses. Therefore, we utilize these predicted frame-wise gloss probabilities as pseudo-targets to provide additional fine-grained supervision, complementing the coarse-grained CTC loss. In line with our multi-modality design, we take the averaged gloss probabilities derived from the four head networks as pseudo-targets to guide the learning process of each stream. Specifically, we minimize the KL-divergence between these pseudo-targets and the predictions obtained from the four head networks. This loss, which we refer to as frame-level self-distillation loss, serves a dual purpose. Firstly, it provides frame-level supervision, enabling more precise guidance during training. Secondly, it facilitates knowledge distillation, as the late ensemble's knowledge is distilled back into each individual stream. By applying the frame-level self-distillation loss, our model benefits from both fine-grained supervision and knowledge distillation, enhancing the learning process and overall performance of each stream within the multi-modal architecture.

