# OpenReview forum: "SignKD: Multi-modal Hierarchical Knowledge Distillation for Continuous Sign Language Recognition"
_ICLR.cc/2024/Conference — ICLR 2024 Conference Withdrawn Submission_

### Official Review · Reviewer_eEWQ · 2023-10-27

**Soundness:** 3 good
**Presentation:** 3 good
**Contribution:** 2 fair
**Rating:** 5
**Confidence:** 4

**Summary:**

This paper introduces a multi-modal model and a concept of hierarchical knowledge distillation (HKD) to tackle the Continuous Sign Language Recognition (CSLR) task. Specifically, first, the multi-modal model takes RGB sign language video and the corresponding pose and optical flow obtained from off-the-shelf detectors. Each modality is processed with a separate feature encoder, and MLP-based fusion layer enables information exchange between different modalities in the intermediate stages of S3D architecture. Secondly, HKD mechanism transfers the multi-modal knowledge from the multi-modal teacher model to a student model that only takes RGB video as input. Additionally combined with Sign pyramid networks (SPN) and self-distillation (SS) technique, the final model achieves strong CSLR performances in different benchmarks.

**Strengths:**

- Transferring multi-modal knowledge learned by multiple streams to a single-modality network via knowledge distillation seems to be an innovative and practical idea in the CSLR community. The advantage of this method is that while enjoying multi-modal knowledge from the teacher model, the student model itself does not need to take additional modality inputs such as optical flow and pose, where such extra modalities actually incur computational overhead during inference time.

- The presentation for the effect of each proposed component is quite clear from both teacher-side and student-side. For the teacher side, improvements are gradually observed when more modalities are combined, and more fusion layers are applied (Table 3). From the student-side, similar to the teacher model, the performance increases when the knowledge distillation loss terms are applied in feature levels and the output-level (Table 4b).

- Compared to the provided previous CSLR methods, the proposed scheme achieves strong results in both cases of multi-modal network and single-modal network in some standard CSLR benchmarks.

**Weaknesses:**

[Major]
- Although the overall flow of the technical design is plausible, the proposed SignKD framework combines two existing components well-established in literature (i.e., multi-modal fusion layer and feature-level knowledge distillation), which is not an astounding result and can weaken the technical novelty of the work. As submitted to a top-tier ML conference, further theoretical or intuitive insight on the architectural design would be expected.

- The claim for the state-of-the-art performance in CSLR tasks needs to be reconsidered. Specifically for the video-only methods, is the video-based backbone S3D pretrained on video data, or initialized from scratch? While the most performant baseline SEN (Hu et al., 2023c) is based on the architecture with frame-level 2D CNN followed by LSTM, but the proposed work adopts a strong video backbone. Such architectural differences and the following advantages should be acknowledged when comparing performances. Can the proposed work still achieve superior performances even when built upon such frame-level architecture? In addition, for the multi-modal-based experiments, comparison with baseline methods in the same modality condition is missing. For example, how much the performance drop would be observed when the same kind of modalities with baseline methods (i.e., video+pose) are used for training the final multi-modal teacher model?

[Minor]
- For the additional techniques explained in Appendix C. in detail, missing citation for the sign pyramid network [Chen et al., 2022]. In addition, frame-level supervision has been also previously explored [Min et al., 2021, Hao et al., 2021, Jang et al., 2023]. Further discussions on advantages or differences of the proposed work compared to previous works would be more helpful for potential readers.

- Computational aspects for the proposed model are missing. Batch size during training is not provided. In addition, as the training cost for a CSLR model is highly demanding, it would be beneficial to share total training hours and required VRAM memory. Moreover, how many frames the final student model can process during inference (i.e., FPS)? Considering real-world deployment, reporting such quantities and comparing with existing CSLR methods is suggested.

- For the ablation results from Table 4(b), one additional row corresponding to the result where only CTC and O are marked is expected. This can be compared with the item where all the feature-level distillation losses are applied (the last row).

- Related to the reproducibility, it is unclear whether the codes for training and inference would be released. If there is no plan to open the codes, could you provide a brief reason?

**Questions:**

[Questions, discussions and suggestions]

- Considering current standard CSLR benchmarks such as PHOENIX-2014, in my opinion, it would be relatively easy for a model to capture motion information of a signer since it has uniform background and the only component with motion is caused by a human. Related to this, it would be interesting to see the stress-testing result for the already trained model on challenging environments [Jang et al., 2022].

- Presenting sample-level qualitative visualizations such as feature activation map for a signer or gloss-level predictions on some video sequences is suggested, as usually done in CSLR literature. It would be valuable to explore whether such visualizations show different characteristics depending on the single-modal (student) network and multi-modal (teacher) network.

- The proposed model includes a lot of loss terms with each weighting factor. Are those weight values sensitive to different datasets?

- Inconsistent notations for Table. (i.e., ‘table 3’ in the first line of section 4.3).

- Typo: ‘is feed’ in the first line of page 4 below the figure.

- What is the meaning of a ‘linear transitional layer’? Is it a standard term?

---

[Reason for the score recommendation]

This paper has distinctive merit as stated in the Strengths section. However, it is quite unsurprising that the combination existing techniques such as multi-modal fusion and knowledge distillation mechanism can produce strong performance. In addition, it is not sufficient to claim as SOTA at this version. It needs to be more rigorous to compare with previous baselines beyond considering just error rates. I would be more inclined to the positive side when those two main issues are addressed during the rebuttal period.

---
References

[Chen et al., 2022] Two-Stream Network for Sign Language Recognition and Translation, NeurIPS 2022.

[Min et al., 2021] Visual Alignment Constraint for Continuous Sign Language Recognition, ICCV 2021.

[Hao et al., 2021] Self-Mutual Distillation Learning for Continuous Sign Language Recognition, ICCV 2021.

[Jang et al., 2023] Self-Sufficient Framework for Continuous Sign Language Recognition, ICASSP 2023.

[Jang et al., 2022] Signing Outside the Studio: Benchmarking Background Robustness for Continuous Sign Language Recognition, BMVC 2022.

---

### Official Review · Reviewer_KJqg · 2023-10-29

**Soundness:** 3 good
**Presentation:** 3 good
**Contribution:** 2 fair
**Rating:** 3
**Confidence:** 5

**Summary:**

The paper first proposes a strong teacher model which jointly models video, keypoints, and optical flow for continuous sign language recognition. Besides, the paper also proposes SignKD, a hierarchical knowledge distillation techinique to transfer the knowledge from the strong teacher model to a single-modality (video-based) student model. The overall system achieves SOTA performance on three widely adopted benchmarks.

**Strengths:**

1. Knowledge distillation in sign language recognition is relatively under-explored.
2. Strong performance on three benchmarks.
3. Detailed ablation studies.

**Weaknesses:**

1. My major concern is the novelty. Since TwoStream Network (neurips 2022) already proposed to jointly model videos and keypoints, adding another modality, i.e., optical flow, is somehow incremetal. Many techiniques are directly borrowed from TwoSream Network, including how to represent keypoints, joint head network, and sign pyramid network. Also, jointly modeling the three modalities has already appeared in a relevant literature [1] (although [1] is for isolated recognition, it is trivial to extend to continuous recognition by just replacing the cross-entropy loss with the CTC loss), and thus the authors' claim that they are the first to utilize the three modalities is weak. The DNF cited in the paper also proposes to model videos and optical flow.

2. Compare to some purely video-based methods (without distillation), e.g., CorrNet and CVT-SLR, the performance gain is small as shown in Table 1.

3. Given that it is a empirical paper, similar to TwoStream Network, more comprehensive evaluation including sign language translation shall be considered.

4. What does "hierarchical" mean? The authors distill features in a stage-to-stage way. Then where is "hierarchical"? The authors also claim that "this involves transferring knowledge from the shallow layers to the deep layers" in page 6, but I cannot find which techinique is corresponding to this claim.

5. Why the authors choose a somehow old algorithm to extract optical flow? Some new methods, e.g., RAFT, shall be considered.

Generally, only the distillation part of the paper is relatively new, but I don't think it is enough for ICLR. As a more empirical paper, I suggest the authors to further evaluate their method on sign language translation.

[1] Skeleton Aware Multi-modal Sign Language Recognition, CVPRW 2021

**Questions:**

See weakness.

---

### Official Review · Reviewer_7ffA · 2023-11-02

**Soundness:** 2 fair
**Presentation:** 2 fair
**Contribution:** 2 fair
**Rating:** 3
**Confidence:** 5

**Summary:**

This paper proposes a multi-modal framework to leverage RGB videos, keypoints and optical flow modalities. Moreover, it investigates different fusion strategies to combine different information and introduces a hierarchical knowledge distillation  technique to enhance single-modal performance. The results in public benchmarks validate the effectiveness of the proposed framework.

**Strengths:**

1. The overall contribution of this paper is clear and easy to understand.
2. The different modalities, i.e., RGB video, keypoints and optical flow contains different useful information for continuous sign language recognition.  This paper explores the effectiveness of these information and combines them into a unified framework.
3. This paper explores the distillation technique between the multi-modal framework and single-modal framework. The proposed distillation loss function appears to be reasonable.

**Weaknesses:**

1. The core idea is simple and limited novel.  The utilization of different modalities is widely validated in many general computer vision tasks, i.e., human action recognition, action temporal localization, etc.  Directly stacking multiple modalities to enhance network performance makes me question the novelty of the paper.  The author should explain their contributions in more detail.
2. The proposed fusion strategy includes MLP, Attention and Convolution. The results in Table 3 reveals the superiority of using MLP. However,  the author did not analyze why MLP outperforms the other two fusion methods, but merely compared them in terms of performance. I suggest that a more in-depth discussion would help reviewers better understand your method.
3. The paper claims that "By incorporating video, keypoints and optical flow, our model can effectively capture movement patterns and hand gestures ". However,  I cannot find any experiment or visualization results to support this point.
4. The paper claims the distillation technique could maintain the high performance while reducing the computational cost and resource requirements. I believe that a comparison of training time, parameter count, and data preprocessing time between this method and other methods should be provided to support this point. From my viewpoint,  keypoint extraction and optical flow computation consume much time.
5. The proposed sign pyramid network is similar to the architecture in the TwoStream-SLR [chen et al., 2022b]. I suggest the author rigorously explains the differences or cites this paper in the proper position.

**Questions:**

See the weaknesses part above.